# Identification of Nano-Metal Oxides That Can Be Synthesized by Precipitation-Calcination Method Reacting Their Chloride Solutions with NaOH Solution and Their Application for Carbon Dioxide Capture from Air—A Thermodynamic Analysis

**DOI:** 10.3390/ma16020776

**Published:** 2023-01-12

**Authors:** Ei Ei Khine, George Kaptay

**Affiliations:** 1Institute of Physical Metallurgy, Metal Forming and Nanotechnology, University of Miskolc, 3515 Miskolc, Hungary; 2ELKH-ME Materials Science Research Group, University of Miskolc, 3515 Miskolc, Hungary

**Keywords:** nano-oxides, synthesis, thermodynamic analysis, carbon capture

## Abstract

Several metal oxide nanoparticles (NPs) were already obtained by mixing NaOH solution with chloride solution of the corresponding metal to form metal hydroxide or oxide precipitates and wash—dry—calcine the latter. However, the complete list of metal oxide NPs is missing with which this technology works well. The aim of this study was to fill this knowledge gap and to provide a full list of possible metals for which this technology probably works well. Our methodology was chemical thermodynamics, analyzing solubilities of metal chlorides, metal oxides and metal hydroxides in water and also standard molar Gibbs energy changes accompanying the following: (i) the reaction between metal chlorides and NaOH; (ii) the dissociation reaction of metal hydroxides into metal oxide and water vapor and (iii) the reaction between metal oxides and gaseous carbon dioxide to form metal carbonates. The major result of this paper is that the following metal-oxide NPs can be produced by the above technology from the corresponding metal chlorides: Al_2_O_3_, BeO, CaO, CdO, CoO, CuO, FeO, Fe_2_O_3_, In_2_O_3_, La_2_O_3_, MgO, MnO, Nd_2_O_3_, NiO, Pr_2_O_3_, Sb_2_O_3_, Sm_2_O_3_, SnO, Y_2_O_3_ and ZnO. From the analysis of the literature, the following nine nano-oxides have been already obtained experimentally with this technology: CaO, CdO, Co_3_O_4_, CuO, Fe_2_O_3_, NiO, MgO, SnO_2_ and ZnO (note: Co_3_O_4_ and SnO_2_ were obtained under oxidizing conditions during calcination in air). Thus, it is predicted here that the following nano-oxides can be potentially synthesized with this technology in the future: Al_2_O_3_, BeO, In_2_O_3_, La_2_O_3_, MnO, Nd_2_O_3_, Pr_2_O_3_, Sb_2_O_3_, Sm_2_O_3_ and Y_2_O_3_. The secondary result is that among the above 20 nano-oxides, the following five nano-oxides are able to capture carbon dioxide from air at least down to 42 ppm residual CO_2_-content, i.e., decreasing the current level of 420 ppm of CO_2_ in the Earth’s atmosphere at least tenfold: CaO, MnO, MgO, CdO, CoO. The tertiary result is that by mixing the AuCl_3_ solution with NaOH solution, Au nano-particles will precipitate without forming Au-oxide NPs. The results are significant for the synthesis of metal nano-oxide particles and for capturing carbon dioxide from air.

## 1. Introduction

Metal oxide nanoparticles have been produced from different reagents by different synthesis techniques, including precipitation [1], combustion [2], sol-gel [3], microwave [4], mechanochemical [5], and hydrolysis [6]. Metal oxides nanoparticles can be used as catalysts or toxic-waste remediation agents for their activity, biological toxicity in pharmaceuticals, or as additives in refractory and paint industries, etc.

Al_2_O_3_ nanoparticles were synthesized using the precipitation technique from various reagents, including aluminum isopropoxide, Al(NO_3_)_3_·9H_2_O and AlCl_3_·6H_2_O [7] as well as additional reactants, such as HCl, deionized water, ethyl alcohol, ammonium hydroxide, and ethanol [8].

BeO nanopowder was produced using the hydrothermal method [9], the sol-gel method [10], and precipitation [11,12] from beryllium hydroxide, beryllium sulfate tetrahydrate and ammonium hydroxide.

CaO has been produced by using different synthesis methods such as precipitation [13,14], thermal decomposition [15], sol-gel [16] using different reagents such as CaCO_3_, Ca(NO_3_)_2_, CaCl_2_, and NaOH.

CdO has been produced using cadmium acetate dihydrate Cd(COOCH)_3_, Cd(NO_3_)_3_·4H_2_O and NaOH by the precipitation method [17,18,19], photochemical synthesis [20] and the soft chemical method [21]. Furthermore, CdO can be formed via the green chemistry process [22] and hydrothermal synthesis [23].

CoO nanoparticles have been produced using cobalt nitrate, cobalt sulfate, cobalt chloride and cobalt acetate [24] only at very high temperatures, and the produced nanoparticles are only stable in the un-oxidizing environment, while CoO transformed into Co_3_O_4_ in air [25]. Co_3_O_4_ has been produced through different routes such as green synthesis, precipitation, wet chemical, template-assisted, sol-gel, hydrothermal or solvothermal, nonchemical and chemical bath deposition methods [26].

CuO nanowires, nanorods, and nano pellets have been mostly produced from copper chloride and NaOH by using poly templating [27], precipitation [28], hydrothermal, and sol-gel methods [29].

Fe_2_O_3_ nanoparticles in different shapes (nano-plates, nano-sheets) have been produced by using the spray pyrolysis technique [30], photo-assisted electrocatalytic methanol [31], two-step hydrothermal synthesis [32] and microwave hydrothermal synthesis [33]. Moreover, the precipitation method, sonolysis synthesis using FeCl_3_ and NaOH, gas decomposition, sol-gel, bulk solution, and microemulsion are also mentioned as the synthesis methods of iron oxide nanoparticles [34]. Fe_2_O_3_ nanoparticles have been produced through the precipitation method from ferric sulfate precursor and ammonium hydroxide [35].

MgO nanoparticles were produced by using different routes such as green synthesis [36,37], the precipitation method [38,39,40,41] from MgCl_2_ and Mg(NO_3_)_2_, NaOH, the sol-gel method [42], and ammonia solution and the solvent mixed spray pyrolysis technique [43].

MnO nanoparticles with different shapes, such as nanocrystals, nanofibers, nanosheets, nanoclusters, nanocubes and nanoflakes have been produced by using different routes, such as the precipitation method [44], thermal decomposition from Mn(HCOO)_2_ [45] and the sol-gel method from manganese oleate complex [46].

NiO nanosheets have been produced from NiCl_2_ and NaOH using thermal decomposition [47], biosynthesis [48,49] and the microemulsion method [50]. Furthermore, precipitation [51], temperature programmed decomposition (TPD) [52], solid-state decomposition [53] and the sonochemical method [54] were also applied. NiO has been produced through the precipitation technique from NiCl_2_ and urea [55].

SnO nanosheets have been produced by using isothermal synthesis [56], two step-chemical method from SnCl_2_ and deionized water [57], microwave-assisted-hydrothermal synthesis [58] and hydrothermal synthesis [59,60] from SnCl_2_ and ammonium hydroxide.

ZnO nanoparticles can be produced by using different routes such as the hydrothermal process from Zn(NO_3_)_2_, the sol-gel method, the solvothermal method, the microwave-assisted method, the precipitation method, the ultrasonic method, the biological method, and the green synthesis method. ZnO has been produced from different precursors such as Zn(CH_3_CO_2_)_2_·2H_2_O, ZnCl_2_ and Zn(NO_3_)_2_ with different reactants, such as ethanol, oxalic acid, methanol, acetone, NaOH, and butyl alcohol [61].

Table 1 is a summary of nano-oxide synthesis from metal chlorides and NaOH solutions as reagents. One can see that altogether, nine different metal oxide nano-particles have been synthesized so far experimentally using these two types of reagents. One of the goals of this paper is to analyze theoretically, which other nano-oxides can be synthesized in a similar way.

Carbon dioxide CO_2_ is a major contributor to the greenhouse effect. Most CO_2_ emission is due to the combustion of fossil fuel, especially from coal power plants and industrial processes. Metal oxides nanoparticles have been used for the capture of carbon dioxide from air. FeO, Fe_2_O_3_ and Fe_3_O_4_ iron oxides have been investigated for CO_2_ capturing processes [62]. KO_2_ transforms to K_2_CO_3_ in a water and CO_2_ environment [63]. BeO nanotube has been used for CO_2_ absorption [64,65]. Li_2_O transforms to Li_2_CO_3_ under a CO_2_ environment at low temperatures at 600 °C [66]. MgO particles have been produced for carbonation at room temperature to 316 °C and decarbonation process at 320–460 °C [67]. Na_2_O has a better CO_2_ absorption ability than Li_2_O and K_2_O [68]. The second goal of this paper is to find the list of possible metal oxide nano-particles that (i) can be produced by the above explained method and at the same time, (ii) can also absorb carbon dioxide from air efficiently, i.e., reducing the current CO_2_ level of air at least by ten times.

**Table 1 materials-16-00776-t001:** Summary table for the synthesis of metal oxide nanoparticles using metal chloride solutions and NaOH solution as reagents.

Composition of Products	Reagents		Crystallite Size (nm)	Particle Size (nm)	Ref.
CaO	CaCl_2_	NaOH	48	150–200	[69,70,71]
CdO	CdCl_2_	NaOH	40	200	[72]
CdO	CdCl_2_	NaOH	11–24	70	[73]
CoO	CoCl_2_	NaOH	50	224	[74]
CoO	CoCl_2_	NaOH		650	[75]
CuO	CuCl_2_	NaOH		≈ 200	[76]
CuO	CuCl_2_	NaOH	25–66	-	[77]
CuO	CuCl_2_	NaOH	20	50	[78]
Fe_2_O_3_	FeCl_3_	NaOH	19	12–24	[79]
NiO	NiCl_2_	NaOH		25–120	[80,81,82,83]
MgO	MgCl_2_	NaOH	30–70	150	[84]
SnO_2_	SnCl_2_	NaOH			[85]
ZnO	ZnCl_2_	NaOH	16–22	300–500	[86]

## 2. Theoretical Calculations of the Formation of Metal Oxide Nano-Particles through Precipitation-Calcination Method Reacting Metal Chlorides with NaOH

In this paper the following technology is considered. Take a given volume of a 2 M NaOH water solution. Take the equal volume of a metal chloride solution in water with the equal amounts of Cl^−^ and OH^−^ ions. Mix the two solutions and let a precipitate form. Wash the precipitate in water to dilute the Cl^−^ ions to about zero concentration. Filtrate the precipitate and dry it at least to 400 K. If needed, calcine the dry precipitate at a higher temperature. All calculations in this paper are based on the physico-chemical (such as solubility) data taken from the CRC Handbook of Chemistry and Physics [87] and on thermodynamic (such as standard molar Gibbs energy of formation) data taken from the compilation by Barin [88].

There are several reasons to select NaOH as a reagent with this technology:NaOH is a bulk and cheap reagent.NaOH has a good solubility in water: 100 g/100 g of H_2_O at 25 °C. Using its molar mass of 40.0 g/mol, its solubility in water can be re-calculated to molarities as follows: it changes from 10.5 M at 0 °C to 86.8 M at 100 °C. So, a 2 M NaOH solution is a stable and cheap reagent.One of the products of the reaction of metal chlorides with NaOH is NaCl. From the initial 2 M NaOH solution, a maximum of 1 M NaCl solution is formed during the reaction with the equal volume of metal chloride solution. Its solubility in water is 36 g/100 g of H_2_O at 25 °C. From its molar mass of 58.4 g/mol, its solubility can be re-calculated to 6.10 M at 0 °C and 6.70 M at 100 °C in water. Therefore, the maximum concentration of 1 M NaCl that can form during the technology remains in the solution even if the effects of other chlorides are taken into account.During the reaction, NaOH is converted into NaCl; it can be done efficiently, if the difference between their molar standard Gibbs energies is as negative as possible. The molar standard Gibbs energy of formation of NaCl is −383.9 kJ/mol (300 K) and −374.6 kJ/mol (400 K). The molar standard Gibbs energy of formation of NaOH is −379.5 kJ/mol (300 K) and −363.8 kJ/mol (400 K). Thus, the molar standard Gibbs energy change accompanying the transformation of NaOH into NaCl is −4.4 kJ/mol (300 K) and −10.8 kJ/mol (400 K). This negative standard molar Gibbs energy change will help to convert metal chlorides into their hydroxides or oxides.

### 2.1. Condition I Existence and Sufficient Solubility of Stable Chloride for Given Metal

The reaction takes place between an aqueous solution of the metal chloride (MCl_x_) and NaOH. Thus, the first condition is that a given metallic ion should have a stable chloride with a high enough solubility in water. In order to make proper use of the planned 2 M NaOH reagent, the solubility of the metal chloride should be equal or larger than its stoichiometric concentration with this 2 M NaOH considering same volumes of the two solutions. Thus, the required minimum solubility of MCl_x_ is expressed as:(1)$$s_{min,\mathrm{MClx}}=\frac{2}{x}$$
where $s_{\min,\mathrm{MClx}}$ (molarity) is the required minimum solubility of the metal chloride to obey condition I and x is the oxidation number of the metal ion. For M(*x*) = Ca(II) let us substitute the value of *x* = 2 into Equation (1): $s_{\min,\mathrm{CaCl}2}$ = 2/2 = 1 M is obtained. Thus, the solubility of CaCl_2_ in water should be higher than 1 M to obey condition I. The actual solubility of CaCl_2_ in water is $S_{\mathrm{CaCl}2}$ = 81.3 g/100 g of H_2_O at 20 °C. This solubility unit should be re-calculated to molarity as:(2)$$s_{\mathrm{MClx}}=10\cdot \frac{S_{\mathrm{MClx}}}{M_{\mathrm{MClx}}}$$
where, $s_{\mathrm{MClx}}$ (molarity = mol/dm^3^) and $S_{\mathrm{MClx}}$ (g/100 g water) are the actual solubilities of the metal chloride in water in two different units, $M_{\mathrm{MClx}}$ (g/mol) is the molar mass of the metal chloride. Note that Equation (2) is approximate, supposing the density of the solution is about 1 g/cm^3^. Substituting $S_{\mathrm{CaCl}2}$ = 81.3 g/100 g of H_2_O and $M_{\mathrm{CaCl}2}$ = 110.98 g/mol into Equation (2) one can obtain: $s_{\mathrm{CaCl}2}$ = 7.33 M. As this value is higher than the minimum required value of $s_{\min,\mathrm{CaCl}2}$ = 1 M found above, CaCl_2_ obeys condition I, stated as: a metal ion obeys condition I if it has a stable chloride with higher solubility in water (denoted as $s_{\mathrm{MClx}}$ and calculated by Equation (2) compared to its minimum required solubility (denoted as $s_{min,\mathrm{MClx}}$ and calculated by Equation (1). In other words, condition I is obeyed if $s_{\mathrm{MClx}}\geq s_{\min,\mathrm{MClx}}$. If this condition is not obeyed, fast precipitation of metal hydroxide/oxide would be not possible in a reasonable amount when equal volumes of 2 M NaOH solution is mixed with a stoichiometic concentration of metal chloride solution. The metal ions that obey this condition I are listed in the first part of Table 2. Other metal ions that do not obey condition I (i.e., for which $s_{\mathrm{MClx}}<s_{\min,\mathrm{MClx}}$) are listed in the second part of Table 2.

As follows from Table 2, metal chlorides that have sufficient solubility for this technology are made of the following metal ions: M(*x*) = Al(III), Au(III), Ba(II), Be(II), Ca(II), Cd(II), Co(II), Cs(I), Cu(II), Fe(II), Fe(III), In(III), K(I), La(III), Li(I), Mg(II), Mn(II), Na(I), Nd(III), Ni(II), Pr(III), Pt(IV), Rb(I), Sb(III), Sm(III), Sn(II), Sr(II), Y(III), Zn(II). These are altogether 29 metal chlorides of 28 metals. The metal ions that do not obey condition I are: M(*x*) = Ag(I), Au(I), Cu(I), Hg(I), Hg(II), Pb(II), Ra(II), Tl(I) These latter metal chlorides are excluded from further consideration.

### 2.2. Condition II Spontaneous Reaction between Metal Chloride and NaOH

One of the possible reactions of metal chloride with NaOH with the 1:1 molar ratio of chlorine and hydroxide ions leads to the formation of metal hydroxide and NaCl, the latter being fully soluble in the mixed water solution:(3)$$MCl_{x}+x\cdot NaOH=M{\left(OH\right)}_{x}+x\cdot NaCl$$

However, the same reagents in the same molar ratio can lead also to the following reaction with the formation of a metal oxide with NaCl and water, the latter two being fully soluble in the mixed water solution:(4)$${\mathrm{MCl}}_{x}+x\cdot \mathrm{NaOH}={\mathrm{MO}}_{x/2}+x\cdot \mathrm{NaCl}+\frac{x}{2}\cdot H_{2}O\;\mathrm{if}\;x=\mathrm{even}\;(2,\;4,\;\mathrm{etc}.)$$
(5)$${\mathrm{MCl}}_{x}+x\cdot \mathrm{NaOH}=\frac{1}{2}\cdot M_{2}O_{x}+x\cdot \mathrm{NaCl}+\frac{x}{2}\cdot H_{2}O\;\mathrm{if}\;x=\mathrm{odd}\;(1,\;3,\;\mathrm{etc}.)$$

As the left-hand sides of Equation (3) and Equations (4) and (5) are identical, the reaction with a more negative standard molar Gibbs energy change will be preferred by Nature. Moreover, any of reactions (3)–(5) are considered fast (leading to a nano-precipitate) only if they are accompanied by at least −50 kJ/mol-MCl_x_ standard molar Gibbs energy change at 300 K, calculated as:(6)$$\Delta_{r3}G^{0}=x\cdot \left(\Delta_{f}G_{\mathrm{NaCl}}^{0}-\Delta_{f}G_{\mathrm{NaOH}}^{0}\right)+{\Delta G}_{M\left(\mathrm{OH}\right)x}^{0}-\Delta_{f}G_{\mathrm{MClx}}^{0}$$
(7)$$\Delta_{r4a}G^{0}=x\cdot \left(\Delta_{f}G_{\mathrm{NaCl}}^{0}-\Delta_{f}G_{\mathrm{NaOH}}^{0}\right)+\Delta_{f}G_{\mathrm{MOx}/2}^{0}+\frac{x}{2}\cdot \Delta_{f}G_{H2O}^{0}-\Delta_{f}G_{\mathrm{MClx}}^{0}$$
(8)$$\Delta_{r4b}G^{0}=x\cdot \left(\Delta_{f}G_{\mathrm{NaCl}}^{0}-\Delta_{f}G_{\mathrm{NaOH}}^{0}\right)+\frac{1}{2}\cdot \Delta_{f}G_{M2\mathrm{Ox}}^{0}+\frac{x}{2}\cdot \Delta_{f}G_{H2O}^{0}-\Delta_{f}G_{\mathrm{MClx}}^{0}$$
where $\Delta_{f}G_{M\left(\mathrm{OH}\right)x}^{0}$ (J/mol) is the standard molar Gibbs energy of formation of metal hydroxide, $\Delta_{f}G_{\mathrm{MClx}}^{0}$ (J/mol) is the same for metal chloride, $\Delta_{f}G_{\mathrm{NaOH}}^{0}$ (J/mol) is the same for NaOH, $\Delta_{f}G_{\mathrm{NaCl}}^{0}$ (J/mol) is the ssame for NaCl (their difference equals −4.4 kJ/mol at T = 300 K), $\Delta_{f}G_{H2O}^{0}$ (J/mol) is the same for liquid water (=−236.8 kJ/mol at 300 K), $\Delta_{f}G_{\mathrm{MOx}/2}^{0}$ (J/mol) is the same for MO_x/2_ and $\Delta_{f}G_{M2\mathrm{Ox}}^{0}$ (J/mol) is the same for M_2_O_x_. Note that the above requirement (the standard molar Gibbs energy change of reactions (3)–(5) at 300 K should be at least −50 kJ/mol-MCl_x_) is also sufficient to ensure the formation of nano-particles, which are thermodynamically less stable compared to particles larger than 100 nm in diameter [89].

At T = 300 K for M(*x*) = Ca(II): x = 2, $\Delta_{f}G_{\mathrm{CaCl}2}^{0}$ = −747,8 kJ/mol, $\Delta_{f}G_{\mathrm{Ca}\left(\mathrm{OH}\right)2}^{0}\;$ = −897.9 kJ/mol, $\Delta_{f}G_{\mathrm{CaO}}^{0}$ = −603.3 kJ/mol. Substituting these and the above given values into Equations (6) and (7): $\Delta_{r3}G^{0}$ = −158.9 kJ/mol, $\Delta_{r4a}G^{0}$ = −101.1 kJ/mol. There are two conclusions from the latter two values: (i) $\Delta_{r3}G^{0}$ = −158.9 kJ/mol is more negative than $\Delta_{r4a}G^{0}$ = −101.1 kJ/mol and so the formation of Ca(OH)_2_ according to reaction (3) is more probable than the formation of CaO according to reaction (4), (ii) $\Delta_{r3}G^{0}$ = −158.9 kJ/mol is more negative than the above given −50 kJ/mol threshold value, so for M(*x*) = Ca(II) the formation of Ca(OH)_2_ will be preferred by nature. For another 28 metal ions that passed condition I, the results of calculations are given in Table 3.

The following conclusions can be drawn from Table 3:(i).for M(*x*) = Ba(II), Ca(II), Li(I), Mg(II), Sr(II) their hydroxides are preferred by nature, so they passed condition II as hydroxides;(ii).for M(*x*) = Fe(III) its oxide is preferred by nature; the same conclusion can be probably reached for M(x) = In(III), La(III), Mn(II), Nd(III), Ni(II), Pr(III), Sb(III), Sm(III), Sn(II), Y(III) and Zn(II), as the standard molar Gibbs energies of formation of their hydroxides are not given by Barin, so they passed condition II as oxides. Note: there are other literature sources for thermodynamic properties of different hydroxides, but in contrary to the compilation of Barin they do not form a coherent system with the thermodynamic properties of other compounds (chlorides, oxides, etc.), so they are not used here for two reasons: (i) different sources show a too large difference, (ii) most of them lead to the same conclusion that an oxide is preferred;(iii).for M(*x*) = Al(III), Be(II), Cd(II), Co(II), Cu(II), Fe(II) the difference between the standard molar Gibbs energies accompanying reactions (3)–(5) are so small that much probably their mixture is formed, so they passed condition II as a mixture of oxides and hydroxides;(iv).for M(*x*) = Cs(I), K(I), Na(I), Rb(I), Pt(IV) nor their oxide, neither their hydroxide can form, i.e., these five metal ions are excluded from further consideration;(v).M(*x*) = Au(III) is a special case, as by reacting AuCl_3_ and NaOH gold nano-particles precipitate in one step by the reaction: ${\mathrm{AuCl}}_{3}+3\cdot \mathrm{NaOH}=\mathrm{Au}+3\cdot \mathrm{NaCl}+\frac{3}{2}\cdot H_{2}O+\frac{3}{4}\cdot O_{2}\left(g\right)$, accompanied by the standard molar Gibbs energy change of −321.1 kJ/mol, being more negative by −39 kJ/mol compared to reactions (3) and (5), see Table 3 (for experimental proof see [90,91]). That is why M(*x*) = Au(III) is also excluded from further consideration, as it cannot provide oxide nano-particles, mostly because Au_2_O_3_ is the only oxide in Table 3 that has positive standard molar Gibbs energy of formation. Finally, conditions I-II are obeyed only by 29 − 5 − 1 = 23 metal ions of 22 metals.

### 2.3. Condition III Fast Precipitation of Metal Hydroxides or Metal Oxides

The metal hydroxide (or oxide) formed in the previous step should precipitate fast in order to form nano-crystals. For that, the solubility of metal hydroxide (or oxide) in water should be much lower than its actual concentration in the water solution. The maximum concentration of metal hydroxide is about 1/*x* M, supposing that the same volume of NaOH solution was added to the same volume of the metal chloride solution with concentrations discussed above. To make sure the precipitation is fast, we need at least ten times less solubility compared to the maximum concentration, i.e.,
(9)$$s_{\max,M\left(\mathrm{OH}\right)x}=\frac{0.1}{x}$$
where, $s_{\max,M\left(\mathrm{OH}\right)x}$ (molarity) is the maimum allowed solubility of the metal hydroxides to satisfy condition III. Substituting *x* = 2 into Equation (9) for M(*x*) = Ca(II), the maximum solubility of Ca(OH)_2_ is found as $s_{\max,\mathrm{Ca}\left(\mathrm{OH}\right)2}$ = 0.05 M in water. The actual solubility is $S_{\mathrm{Ca}\left(\mathrm{OH}\right)2}$ = 0.16 g/100 g water at T = 293 K. This value should be re-calculated to the unit of molarity by an equation being similar to Equation (2):(10)$$s_{M\left(\mathrm{OH}\right)x}=10\cdot \frac{S_{M\left(\mathrm{OH}\right)x}}{M_{M\left(\mathrm{OH}\right)x}}$$
where $s_{M\left(\mathrm{OH}\right)x}$ (molarity) and $S_{M\left(\mathrm{OH}\right)x}$ (g/100 g water) are the solubilities of the metal hydroxide in two different units, while $M_{M\left(\mathrm{OH}\right)x}$ (g/mol) is the molar mass of the metal hydroxide. Substituting $S_{\mathrm{Ca}\left(\mathrm{OH}\right)2}$ = 0.16 g/100 g and $M_{\mathrm{Ca}\left(\mathrm{OH}\right)2}$ = 74.1 g/mol into Equation (10): $s_{M\left(\mathrm{OH}\right)x}$ = 0.022 is found. As this value is smaller than the maximum allowed value of $s_{\max,M\left(\mathrm{OH}\right)x}$ = 0.05 M, so M(*x*) = Ca(II) satisfies condition III. The same calculation is repeated in Table 4 for those metal ions that satisfied both conditions I and II above.

The metal ions satisfying conditions I-II but not obeying condition III are listed at the bottom of Table 4: M(*x*) = Ba(II), Li(I), Sr(II). These metal ions are excluded from further consideration. As follows from Table 4, all other metal ions that passed conditions I-II also pass condition III. As was mentioned above, some of the metal ions that passed conditions I-II preferably form oxides. As the solubility of all those oxides are negligible in water, all those ions pass condition III. Summarizing, altogether 20 metal ions of 19 metals pass conditions I-II-III:metal ions M(*x*) = Ca(II) and Mg(II) pass conditions I-II-III as hydroxides;metal ions M(*x*) = Fe(III), In(III), La(III), Mn(II), Nd(III), Ni(II), Pr(III), Sb(III), Sm(III), Sn(II), Y(III) and Zn(II) pass conditions I-II-III as oxides;metal ions M(*x*) = Al(III), Be(II), Cd(II), Co(II), Cu(II) and Fe(II) pass conditions I-II-III as a mixture of their oxides and hydroxides.

The above precipitates are dispersed in a water solution and can be contaminated by dissolved NaCl. That is why they should be washed by water to remove NaCl, so when the precipitates are dried, NaCl would not co-precipitate and would not contaminate them.

Those metal ions that passed the above conditions I-II-III as oxides, should be dried only, preferably at around 400 K, i.e., somewhat above the boiling point of water. As a result, oxide nano-particles can be obtained for the following 12 metal ions M(*x*) = Fe(III), In(III), La(III), Mn(II), Nd(III), Ni(II), Pr(III), Sb(III), Sm(III), Sn(II), Y(III) and Zn(II). As follows from the above, there are further six metal ions that are precipitated at least partly as oxides: M(x) = Al(III), Be(II), Cd(II), Co(II), Cu(II) and Fe(II). For the oxide part of the precipitates from these six metal ions it is also sufficient to dry the system at around 400 K to obtain the desired nano-particles. However, the hydroxide part of those six metal ions should be calcined to obtain oxide nano-particles. The same is true for the two metal ions that are precipitated in the form of hydroxides: M(*x*) = Ca(II) and Mg(II). Therefore, for these eight metal ions, condition IV is considered below.

### 2.4. Condition IV the Ability of Metal Hydroxides to Convert into Metal Oxides upon Calcination at a Reasonably Low Temperature

The following chemical reactions are expected upon heating a metal hydroxide:(11)$$M{\left(\mathrm{OH}\right)}_{x}={\mathrm{MO}}_{x/2}+\frac{x}{2}\cdot H_{2}O\left(g\right)\to \;\mathrm{if}\;x=\mathrm{even}\;(2,\;4,\;\mathrm{etc}.)$$
(12)$$M{\left(\mathrm{OH}\right)}_{x}=\frac{1}{2}\cdot M_{2}O_{x}+\frac{x}{2}\cdot H_{2}O\left(g\right)\to \;\mathrm{if}\;x=\mathrm{odd}\;(1,\;3,\;\mathrm{etc}.)$$

For example, for M(*x*) = Ca(II) and *x* = 2 Equation (11) is simplified to: $Ca{\left(OH\right)}_{2}=CaO+H_{2}O$. As the only gaseous component of reactions (11) and (12) is on their right-hand sides (water vapor), these reactions will be shifted to the right with increasing temperature. Therefore, it is expected that all metal hydroxides will thermally decompose at a sufficiently high temperature. Those metal ions will pass condition IV, for which the normal decomposition temperature is much below the melting points of their oxides and hydroxides, and so after calcination the nano-size of the oxide particles will be similar to the nano-size of the hydroxide particles before calcination.

The normal decomposition temperature will be the temperature at which the partial pressure of water vapor reaches 1 bar, i.e., when the standard molar Gibbs energies of reactions (11) and (12) will be zero (at a lower temperature this will be positive, while at a higher temperature this will be negative). This is demonstrated in Figure 1 for M(*x*) = Ca(II). As follows from Figure 1, the normal decomposition temperature of Ca(OH)_2_ to CaO is found at about T_d_ = 785 K.

Data obtained in a similar way as shown in Figure 1 are collected in Table 5 for other metal hydroxides. After Ca(OH)_2_ the second highest decomposition temperature is obtained for Mg(OH)_2_, being equal to 542 K. As follows from Table 5, the decomposition temperatures of all other hydroxides are in the interval of 300 … 400 K. Moreover, all the decomposition temperatures found in Table 5 are much below the corresponding melting points of the hydroxides and oxides of the same metals. Therefore, we can conclude that nano-oxide particles can be obtained by calcination of the hydroxides of all the eight metal ions of Table 5 at reasonably low calcination temperatures: M(*x*) = Al(III), Be(II), Ca(II), Cd(II), Co(II), Cu(II), Fe(II) and Mg(II). Thus, all these eight metal ions obey condition IV. Note: calcination should be performed at least at 400 K (or above) even for metal hydroxides with lower decomposition temperatures to make sure that dry oxide nano-particles are obtained.

These eight metal ions together with the above mentioned 12 metal ions are those 20 metal ions that obey all conditions I-II-III-IV: M(*x*) = Al(III), Be(II), Ca(II), Cd(II), Co(II), Cu(II), Fe(II), Fe(III), In(III), La(III), Mg(II), Mn(II), Nd(III), Ni(II), Pr(III), Sb(III), Sm(III), Sn(II), Y(III) and Zn(II). Thus, the chloride solutions of these metal ions mixed with a NaOH solution leads to oxide or hydroxide precipitates that lead to the formation of oxide nano-particles upon heating to at least 400 K to dry the system or to a somewhat higher temperature for calcination. These findings are confirmed by Table 1.

### 2.5. Additional Condition V: Ability of Metal Oxides to Capture CO_2_ from Air

All the above is performed in order to produce oxide nano-particles. One of their possible applications is to capture CO_2_ from air, according to the reactions:(13)$${\mathrm{MO}}_{x/2}+\frac{x}{2}\cdot {\mathrm{CO}}_{2}\left(g\right)=M{\left({\mathrm{CO}}_{3}\right)}_{x/2}\;\mathrm{if}\;x=\mathrm{even}\;(2,\;4,\;\mathrm{etc}.)$$
(14)$$\frac{1}{2}\cdot M_{2}O_{x}+\frac{x}{2}\cdot {\mathrm{CO}}_{2}\left(g\right)=\frac{1}{2}\cdot M_{2}{\left({\mathrm{CO}}_{3}\right)}_{x}\;\mathrm{if}\;x=\mathrm{odd}\;(1,\;3,\;\mathrm{etc}.)$$

As in reactions (13) and (14), the only gaseous component (CO_2_) is on the left-hand side, the reactions are shifted to the left with increasing T. Thus, the possibility of these reactions will be checked at T = 300 K. If they do not work at T = 300 K, they will not work either at higher temperatures.

The current concentration of CO_2_ in the atmosphere is 420 ppm. To call a capturing technology efficient, it should decrease this value at least by ten times. Thus, reactions (13) and (14) should have an equilibrium at a partial pressure of CO_2_ of 42 ppm or below, i.e., p_CO2_ ≤ 4.2 × 10^−5^ bar for 1 bar of air pressure. Thus, the equilibrium constant of reactions (13) and (14) should be at least the inverse of this value to the power of *x*/2: (2.38 × 10^4^)*^x^*^/2^ at T = 300 K. For M(*x*) = Ca(II): *x* = 2, i.e., K_min_ should be at least 2.38 × 10^4^ at T = 300 K. This is possible if the molar standard Gibbs energy change accompanying reaction (13) is more negative than –RTlnK_min_ = −25.1 kJ/mol. The standard molar Gibbs energies of formation at T = 300 K for the case of M(*x*) = Ca(II): $\Delta_{f}G_{\mathrm{CaO}}^{0}$ = −603.3 kJ/mol, $\Delta_{f}G_{\mathrm{CO}2}^{0}\;$ = −394.4 kJ/mol, $\Delta_{f}G_{\mathrm{CaCO}3}^{0}$ = −1127.3 kJ/mol. Thus, the standard molar Gibbs energy change accompanying reaction (13) for M(*x*) = Ca(II), i.e., that of reaction $CaO+CO_{2}=CaCO_{3}$ is −129.6 kJ/mol. As this value is more negative than the value of –RTlnK_min_ = −25.1 kJ/mol, one can conclude that CaO nano-particles are able to capture CO_2_ from air at T = 300 K below the level of 42 ppm, i.e., M(*x*) = Ca(II) obeys condition V. Similar calculations are performed for another 19 M(*x*) ions in Table 6. Note that the thermodynamic instability of the nano-sized oxide particles will move reactions (13) and (14) even somewhat further to the right compared to the results of Table 6 [89].

As one can see from Table 6, among the 20 metal nano-oxides that can be produced by the technology described above, the following five oxides are able to capture carbon dioxide from air at least down to 42 ppm residual CO_2_-content (from strongest to weakest absorber): M(*x*) = Ca(II), Mn(II), Mg(II), Cd(II), Co(II). Additionally, the following three oxides can also capture some CO_2_, but not down to 42 ppm: M(*x*) = Fe(II), Zn(II), Ni(II).

The above is confirmed for M(x) = Fe(II), Zn(II), La(III), Ce (III) and Sn(II) under 150 bar in [92,93,94,95,96]. Further, CaO [71,97,98,99,100], BeO [64], Co_3_O_4_ [101], MgO [102,103], FeO, Fe_2_O_3_, Fe_3_O_4_ [104], Li_2_O [66,105], CuO [106,107] and NiO [108] were used as sorbents in a CO_2_ environment. ZnO has been also studied for CO_2_ capturing in CO_2_ environment using the pressure at 1–25 bar [109,110,111,112].

## 3. Conclusions

Although many different nano-oxides have been synthesized as shown above, the complete list of possible nano-oxides to be synthesized by the precipitation-calcination route using metal chlorides and NaOH is missing. In this paper, this knowledge gap is filled by a theoretical method. Moreover, the ability of those oxides to capture carbon dioxide from air is discussed.

The way to synthesize oxide nano-particles by mixing the NaOH solution with a metal chloride solution to form metal hydroxide or oxide precipitates and drying or calcining the latter to form metal oxide nanoparticles is theoretically considered in this paper to select suitable metal chlorides.

The chlorides of the following 20 ions were identified as possible candidates for metal oxide nano-particle synthesis: M(*x*) = Al(III), Be(II), Ca(II), Cd(II), Co(II), Cu(II), Fe(II), Fe(III), In(III), La(III), Mg(II), Mn(II), Nd(III), Ni(II), Pr(III), Sb(III), Sm(III), Sn(II), Y(III) and Zn(II). These chlorides lead to the following 20 nano-oxide particles, if during the production the oxidation number of the metal is not changed: Al_2_O_3_, BeO, CaO, CdO, CoO, CuO, FeO, Fe_2_O_3_, In_2_O_3_, La_2_O_3_, MgO, MnO, Nd_2_O_3_, NiO, Pr_2_O_3_, Sb_2_O_3_, Sm_2_O_3_, SnO, Y_2_O_3_ and ZnO. If calcination is performed in oxidizing atmosphere the above oxides can be oxidized further, depending on temperature and oxygen partial pressure.

From the analysis of the literature, the following nine nano-oxides have been obtained so far experimentally by this technology: CaO, CdO, Co_3_O_4_, CuO, Fe_2_O_3_, NiO, MgO, SnO_2_ and ZnO. One can see that the two lists overlap, confirming the validity of our analysis (note: Co_3_O_4_ and SnO_2_ were obtained under oxidizing conditions during calcination in air). None of the nano-oxides not predicted by us have been actually synthesized, further proving the validity of our theoretical analysis. One can also see that Al_2_O_3_, BeO, In_2_O_3_, La_2_O_3_, MnO, Nd_2_O_3_, Pr_2_O_3_, Sb_2_O_3_, Sm_2_O_3_ and Y_2_O_3_ have not yet been synthesized experimentally from our list of theoretically possible candidate nano-oxides. We predict that these nano-oxides will be synthesized in the future with the above technology.

Among the above 20 nano-oxides, the following five nano-oxides are able to capture carbon dioxide from air at least down to 42 ppm residual CO_2_-content (from strongest to weakest absorber): CaO, MnO, MgO, CdO, CoO. The following three nano-oxides are somewhat weaker absorbers of carbon dioxide: FeO, ZnO, NiO.

Let us note that mixing the AuCl_3_ solution with NaOH solution leads to the immediate precipitation of Au nano-particles without forming Au-oxide NPs.

## Figures and Tables

**Figure 1 materials-16-00776-f001:**
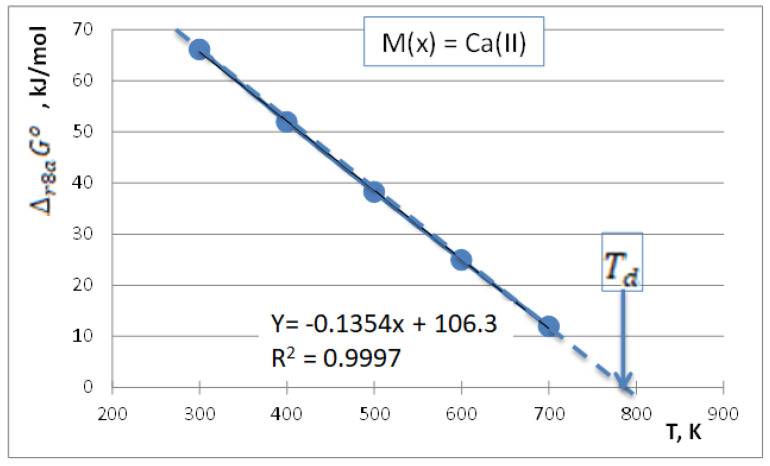
The temperature dependence of the standard molar Gibbs energy change of reaction (11) for M(*x*) = Ca(II). The extrapolated line crosses zero at T_d_ = 785 K.

**Table 2 materials-16-00776-t002:** Metal ions whose chlorides have (or not) high enough solubility to obey condition I.

M	*x*	$s_{min,MClx}$ M	$S_{MClx}$ g/100 g H_2_O	$M_{MClx}$ g/mol	$s_{MClx}$ M	Obey?
Al	3	0.67	45.1	133.33	3.38	Yes
Au	3	0.67	68	303.35	2.24	Yes
Ba	2	1	37	208.2	1.78	Yes
Be	2	1	71.5	79.91	8.95	Yes
Ca	2	1	81.3	110.98	7.33	Yes
Cd	2	1	120	183.3	6.55	Yes
Co	2	1	56.2	129.83	4.33	Yes
Cs	1	2	191	168.35	11.35	Yes
Cu	2	1	75.7	134.45	5.63	Yes
Fe	2	1	65	126.75	5.13	Yes
Fe	3	0.67	91.2	162.2	5.62	Yes
In	3	0.67	195.1	221.15	8.82	Yes
K	1	2	35.5	74.55	4.76	Yes
La	3	0.67	95.7	245.25	3.90	Yes
Li	1	2	84.5	42.39	19.93	Yes
Mg	2	1	56	95.21	5.88	Yes
Mn	2	1	77.3	125.84	6.14	Yes
Na	1	2	36	58.44	6.16	Yes
Nd	3	0.67	100	205.55	4.86	Yes
Ni	2	1	67.5	129.59	5.21	Yes
Pr	3	0.67	96.1	247.25	3.89	Yes
Pt	4	0.5	142	336.9	4.21	Yes
Rb	1	2	93.9	120.92	7.77	Yes
Sb	3	0.67	987	228.15	43.26	Yes
Sm	3	0.67	93.8	256.75	3.65	Yes
Sn	2	1	178	189.6	9.39	Yes
Sr	2	1	54.7	158.52	3.45	Yes
Y	3	0.67	75.1	195.26	3.85	Yes
Zn	2	1	408	136.28	29.94	Yes
Ag	1	2	0.00019	143.32	1.33 × 10^−5^	No
Au	1	2	0.000031	232.42	1.33 × 10^−6^	No
Cu	1	2	0.0047	98.99	4.75 × 10^−4^	No
Hg	1	2	7.31	271.52	0.23	No
Hg	2	1	0.0004	472.09	8.47 × 10^−6^	No
Pb	2	1	1.08	278.1	0.0388	No
Ra	2	1	24.5	296.09	0.827	No
Tl	1	2	0.33	189.68	0.0174	No

**Table 3 materials-16-00776-t003:** Results of calculations by Equations (6)–(8) at T = 300 K (all values are in kJ/mol-MCl_x_).

M	*x*	$\Delta_{f}G_{MClx}^{0}$	$\Delta_{f}G_{M\left(OH\right)x}^{0}$	$\Delta_{f}G_{MOx/2}^{0}$ $\Delta_{f}G_{M2Ox}^{0}$	$\Delta_{r3}G^{0}$	$\Delta_{r4a}G^{0}$ $\Delta_{r4b}G^{0}$	$\Delta_{r4}G^{0}$ $-\Delta_{r3}G^{0}$	Preference
Al	3	−629.5	−1137.8	−1581.7	−521.5	−529.8	−8.3	mixed *
Au	3	−47.3	−316.2	+78.4	(−282.1)	(−281.9)	(+0.2)	Au **
Ba	2	−810.0	−859.0	−525.2	−57.8	+39.2	+97.0	Ba(OH)_2_
Be	2	−449.2	−815.4	−578.9	−375.0	−375.3	−0.3	mixed
Ca	2	−747.8	−897.9	−603.3	−158.9	−101.1	+57.8	Ca(OH)_2_
Cd	2	−343.6	−473.2	−229.1	−138.4	−131.1	+7.3	mixed
Co	2	−269.4	−453.6	−214.1	−193.0	−190.3	+2.7	mixed
Cu	2	−173.5	−358.5	−128.1	−193.8	−200.2	−6.4	mixed
Fe	2	−302.1	−486.5	−251.3	−193.2	−194.8	−1.6	mixed
Fe	3	−333.5	−695.7	−741.8	−375.4	−405.8	−30.4	Fe_2_O_3_
In	3	−461.8	---	−830.0	---	−321.6	---	In_2_O_3_
La	3	−994.9	---	−1705.4	---	−226.2	---	La_2_O_3_
Li	1	−383.9	−438.7	−561.9	−59.2	−20.9	+38.4	LiOH
Mg	2	−591.8	−833.1	−568.7	−250.1	−222.5	+27.6	Mg(OH)_2_
Mn	2	−440.2	---	−362.8	---	−168.4	---	MnO
Nd	3	−966.1	---	−1720.5	---	−262.6	---	Nd_2_O_3_
Ni	2	−258.9	---	−211.4	---	−198.1	---	NiO
Pr	3	−980.3	---	−1719.7	---	−248.0	---	Pr_2_O_3_
Sb	3	−322.1	---	−633.8	---	−363.2	---	Sb_2_O_3_
Sn	2	−286.0	---	−256.6	---	−216.2	---	SnO
Sm	3	−949.7	---	−1736.8	---	−287.1	---	Sm_2_O_3_
Sr	2	−779.7	−880.6	−561.2	−109.7	−27.1	+82.6	Sr(OH)_2_
Y	3	−927.3	---	−1816.1	---	−349.2	---	Y_2_O_3_
Zn	2	−369.1	---	−320.3	---	−196.8	---	ZnO
Cs	1	−414.2	−370.4	−308.2	+39.4	+137.3	+97.9	No
K	1	−408.6	−378.6	−322.5	+25.6	+124.6	+99.0	No
Na	1	−383.9	−379.5	−378.8	0.0	+71.7	+71.7	No
Pt	4	−163.3	---	---	---	---	---	No
Rb	1	−407.6	---	−299.8	---	+134.9	---	No

* mixed = mixture of the oxide and the hydroxide, ** Au nano-particles form, see text.

**Table 4 materials-16-00776-t004:** The metal hydroxides that have low enough solubility in water at T = 293 K, i.e., obey condition III (among those that passed conditions I and II).

M	*x*	$S_{M\left(OH\right)x}$	$M_{M\left(OH\right)x}$	$s_{max,M\left(OH\right)x}$	$s_{M\left(OH\right)x}$	Obey?
		g/100 g H_2_O	g/mol	M	M	
Al	3	0	78	0.033	0	Yes
Be	2	0	43.01	0.050	0	Yes
Ca	2	0.16	74.1	0.050	0.022	Yes
Cd	2	0.00015	146.4	0.050	1.02 × 10^−05^	Yes
Co	2	0	92.9	0.050	0	Yes
Cu	2	0	97.6	0.050	0	Yes
Fe	2	0.000052	89.9	0.050	5.78 × 10^−06^	Yes
Fe	3	0	106.9	0.033	0	Yes
Mg	2	0.00069	58.3	0.050	1.18 × 10^−04^	Yes
Mn	2	0.00034	88.9	0.050	3.82 × 10^−05^	Yes
Ni	2	0.00015	92.7	0.050	1.62 × 10^−05^	Yes
Sb	3	0	172.78	0.033	0	Yes
Sn	2	0	152.7	0.050	0	Yes
Zn	2	0.000042	99.42	0.050	4.22 × 10^−06^	Yes
Ba	2	4.91	171.3	0.050	0.29	No
Li	1	12.5	23.9	0.100	5.23	No
Sr	2	2.25	121.6	0.050	0.19	No

**Table 5 materials-16-00776-t005:** The normal decomposition temperatures of some metal hydroxides into metal oxides.

M	*x*	T_d_ (K)
Al	3	318
Be	2	354
Ca	2	785
Cd	2	387
Co	2	368
Cu	2	313
Fe	2	342
Mg	2	542

**Table 6 materials-16-00776-t006:** Ability of metal oxides to capture CO_2_ from air at T = 300 K at least down to 42 ppm or below (among those oxides that passed conditions I-IV above).

M	*x*	-RTlnK_min_	$\Delta_{f}G_{oxide}^{o}$	$\Delta_{f}G_{carbonate}^{o}$	$\Delta_{r9}G^{0}$	Obey?
		kJ/mol	kJ/mol	kJ/mol	kJ/mol	
Ca	2	−25.1	−603.3	−1127.3	−129.6	Yes
Cd	2	−25.1	−229.1	−670.0	−46.5	Yes
Co	2	−25.1	−214.1	−636.3	−27.8	Yes
Fe	2	−25.1	−251.3	−666.2	−20.5	Weakly *
Mg	2	−25.1	−568.7	−1011.7	−48.6	Yes
Mn	2	−25.1	−362.8	−816.2	−59.0	Yes
Ni	2	−25.1	−211.4	−617.4	−11.6	Weakly
Zn	2	−25.1	−320.3	−731.0	−16.3	Weakly
Al	3	−37.7	−1581.7	---	---	No
Be	2	−25.1	−578.9	---	---	No
Cu	2	−25.1	−128.1	---	---	No
Fe	3	−37.7	−741.8	---	---	No
In	3	−37.7	−830.0	---	---	No
La	3	−37.7	−1705.4	---	---	No
Nd	3	−37.7	−1720.5	---	---	No
Pr	3	−37.7	−1719.7	---	---	No
Sb	3	−37.7	−633.8	---	---	No
Sm	3	−37.7	−1736.8	---	---	No
Sn	2	−25.1	−256.6	---	---	No
Y	3	−37.7	−1816.1	---	---	No

* Weakly = able to absorb some CO_2_ but not able to decrease its level in air to 42 ppm or below at 300 K.

## Data Availability

The data presented in this study are available on request from the corresponding author.

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
