# Peer review of "Identification of Nano-Metal Oxides That Can Be Synthesized by Precipitation-Calcination Method Reacting Their Chloride Solutions with NaOH Solution and Their Application for Carbon Dioxide Capture from Air—A Thermodynamic Analysis"

_materials, 2023, doi:10.3390/ma16020776_

Round 1
Reviewer 1 Report
Ei Ei Khine reported an article with the title “Identification of nano-metal oxides that can be synthesized by precipitation-calcination method reacting their chloride solutions with NaOH solution and their application for carbon dioxide capture from air. A thermodynamic analysis.” I recommend its publication after the following changes:
1. The quality of the language is very low. It seems that the authors used only synonyms to remove plagiarism.
2. The authors should re-arrange and re-write the abstract in the following order (Problem of research, aim of study, remarkable methodology, remarkable results, and significance of the study).
3. The authors should provide a graphical abstract to support the manuscript.
4. Make sure all subscripts and superscripts are in the right position.
5. The authors should move the last part of the introduction section to the conclusions part and also support it with recommendations and future perspectives.
6. Research articles still lacking recent literature authors should read below recent relevant references and consider for citation
https://doi.org/10.1021/acsomega.2c05133, Nanoscale research letters 16 (1), 1-11, 2021,
Author Response
Reply to reviewer 1
Dear Editor and reviewer,
We thank a lot for your careful analysis of our paper. Herewith we submit our corrected version. All the new text is highlighted as yellow in the paper and here.
Sincerely, George Kaptay, corresponding author
General opinion: Ei Ei Khine reported an article with the title “Identification of nano-metal oxides that can be synthesized by precipitation-calcination method reacting their chloride solutions with NaOH solution and their application for carbon dioxide capture from air. A thermodynamic analysis.” I recommend its publication after the following changes:
Comment 1.1. The quality of the language is very low. It seems that the authors used only synonyms to remove plagiarism.
Reply 1.1. Thank you for your comment. We made our best to improve the language of the manuscript.
Comment 1.2. The authors should re-arrange and re-write the abstract in the following order (Problem of research, aim of study, remarkable methodology, remarkable results, and significance of the study).
Reply 1.2. Thank you for your comment. The abstract is now re-written as: “Several metal oxide nanoparticles (NPs) are already obtained by mixing the NaOH solution with the chloride solution of the corresponding metal to form metal hydroxide or oxide precipitates and wash - dry - calcine the latter. However, the complete list of metal oxide NPs is missing for which this technology is probably useful. The aim of this study was to fill this knowledge gap and to provide a full list of possible metals for which this technology is probably suitable. Our methodology was chemical thermodynamics, analyzing solubilities of metal chlorides, metal oxides and metal hydroxides in water and also standard molar Gibbs energy changes accompanying i). the reaction between metal chlorides and NaOH, ii). the dissociation reaction of metal hydroxides into metal oxide and water vapor and iii). reaction between metal oxides and gaseous carbon dioxide to form metal carbonates. The major result of this paper is that the following metal-oxide NPs can be produced by the above technology from the corresponding metal chlorides: Al2O3, BeO, CaO, CdO, CoO, CuO, FeO, Fe2O3, In2O3, La2O3, MgO, MnO, Nd2O3, NiO, Pr2O3, Sb2O3, Sm2O3, SnO, Y2O3 and ZnO. From the analysis of the literature, the following 9 nano-oxides have been obtained so far experimentally by this technology: CaO, CdO, Co3O4, CuO, Fe2O3, NiO, MgO, SnO2 and ZnO (note: Co3O4 and SnO2 were obtained under oxidizing conditions during calcination in air). Thus, it is predicted here that the following nano-oxides can be potentially synthesized by this technology in the future: Al2O3, BeO, In2O3, La2O3, MnO, Nd2O3, Pr2O3, Sb2O3, Sm2O3 and Y2O3. The secondary result is that among the above 20 nano-oxides the following 5 nano-oxides are able to capture carbon dioxide from air at least down to 42 ppm residual CO2-content, i.e. decreasing its current level in the Earths atmosphere of 420 ppm at least tenfold: CaO, MnO, MgO, CdO, CoO. The terciary result is that by mixing the AuCl3 solution with NaOH solution leads to the immediate precipitation of Au nano-particles without forming Au-oxide NPs. The results are significant for i). synthesis of metal nano-oxide particles and for ii). capturing carbon dioxide from air.”
Comment 1.3. The authors should provide a graphical abstract to support the manuscript.
Reply 1.3. Thank you for your comment and suggestion, the following graphical abstract is added to the end of the paper:
Graphical abstract
Metals for which oxide nano-particles can be synthesized when their chloride solutions are reacted with NaOH solution and the formed precipitate is washed, dried and/or calcined.
|
1A |
8A |
||||||||||||||||
|
|
2A |
3A |
4A |
5A |
6A |
7A |
|
||||||||||
|
|
Be |
|
|
|
|
|
|
||||||||||
|
|
Mg |
3B |
4B |
5B |
6B |
7B |
8B |
8B |
8B |
1B |
2B |
Al |
|
|
|
|
|
|
|
Ca |
|
|
|
|
Mn |
Fe |
Co |
Ni |
Cu |
Zn |
|
|
|
|
|
|
|
|
|
Y |
|
|
|
|
|
|
|
|
Cd |
In |
Sn |
Sb |
|
|
|
|
|
|
La |
|
|
|
|
|
|
|
|
|
|
|
|
|
|
|
|
|
|
|
|
|
|
|
|
|
|
|
|
|
|
|
|
|
|
|
|
Pr |
Nd |
|
Sm |
|
|
|
|
|
|
|
|
|
||||
|
|
|
|
|
|
|
|
|
|
|
|
|
|
|
Comment 1.4. Make sure all subscripts and superscripts are in the right position.
Reply 1.4. Thank you for your comment and suggestions. We revised the whole manuscript accordingly.
Comment 1.5. The authors should move the last part of the introduction section to the conclusions part and also support it with recommendations and future perspectives.
Reply 1.5. Thank you for your comment and suggestions. We moved the last part of the introduction section to the conclusions part.
Comment 1.6. Research articles still lacking recent literature authors should read below recent relevant references and consider for citation
https://doi.org/10.1021/acsomega.2c05133, Nanoscale research letters 16 (1), 1-11, 2021,
Reply 1.6. Thank you for your comment and suggestion. We added the reference to the manuscript.
Reviewer 2 Report
Review:
The present work is devoted to summarizing current research on the synthesis of nanosized oxide powders and the possibility of their application for CO2 capture.
The paper can be published after addressing the following comments: In the literature review, it is necessary to reflect the following works on the preparation of nanopowders of ZnO , Al2O3 (doi:10.1134/S0036023618100157 and others), CaO (doi:10.1016/j.ceramint.2021.11.296 and others), MgO (doi:10.1166/asl.2012.2190 and others) by burning concentrated water-carbohydrate salt solutions. It is also worth noting the work on the use of CaO nanopowder for CO2 adsorption. In general, the work is an interesting theoretical study of the properties with possible projections for the use of oxide nanopowders as CO2 adsorbents.
What is the main question addressed by the research?
This work is general in nature. In fact, methods for obtaining nanosized powders of various oxides are considered. However, these methods of obtaining are simply listed, their comparative characteristics are not given. What are the significant shortcomings of the work.
The main idea of the work is to predict the properties of nanopowders obtained by the precipitation of metal chlorides and oxychlorides. However, it is not clear what this choice is based on. As is known, it is very difficult to remove chloride ions from a solution, and their removal during heat treatment leads to coarsening of powder particles. All of the above leads to doubt about the possibility of practical application of the data presented in the work.
Is it relevant and interesting?
The approach is original. Predicting the possibility of obtaining powders with desired properties by the methods of thermodynamic calculations is interesting.
How original is the topic?
The topic, statement of research objectives and results are original.
What does it add to the subject area compared with other published material?
The originality of the approach is the key achievement of this work. The presented approach has drawbacks, because it is not tested for all oxides, but the approach itself is interesting.
Is the paper well written?
In general, the work is well written, however, the introduction should be revised in detail, regarding the remarks described above.
Is the text clear and easy to read?
The text is difficult to read. English needs to be corrected.
Are the conclusions consistent with the evidence and arguments presented?
Yes, the conclusions to the work are consistent with the text of the work.
Do they address the main question posed?
Yes, the findings are consistent with the purpose of the study.
Author Response
Reply to reviewer 2
Dear Editor and reviewer,
We thank a lot for your careful analysis of our paper. Herewith we submit our corrected version. All the new text is highlighted as yellow in the paper and here.
Sincerely, George Kaptay, corresponding author
Comment 2.1. The present work is devoted to summarizing current research on the synthesis of nanosized oxide powders and the possibility of their application for CO2 capture. The paper can be published after addressing the following comments: In the literature review, it is necessary to reflect the following works on the preparation of nanopowders of ZnO , Al2O3 (doi:10.1134/S0036023618100157 and others), CaO (doi:10.1016/ j.ceramint. 2021.11. 296 and others), MgO (doi:10.1166/asl.2012.2190 and others) by burning concentrated water-carbohydrate salt solutions. It is also worth noting the work on the use of CaO nanopowder for CO2 adsorption. In general, the work is an interesting theoretical study of the properties with possible projections for the use of oxide nanopowders as CO2 adsorbents.
Reply 2.1. Thank you for your comment and suggestions. We cited the references in the manuscript.
What is the main question addressed by the research?
This work is general in nature. In fact, methods for obtaining nanosized powders of various oxides are considered. However, these methods of obtaining are simply listed, their comparative characteristics are not given. What are the significant shortcomings of the work.
Comment 2.2. The main idea of the work is to predict the properties of nanopowders obtained by the precipitation of metal chlorides and oxychlorides. However, it is not clear what this choice is based on. As is known, it is very difficult to remove chloride ions from a solution, and their removal during heat treatment leads to coarsening of powder particles. All of the above leads to doubt about the possibility of practical application of the data presented in the work.
Reply 2.2. Thank you for your comment and suggestion. In this article, the process of the producing of metal oxides nanoparticles are followings: metal chloride + NaOH = metal hydroxides(precipitate) + NaCl (washing with distilled water) = metal hydroxides or metal oxides, then metal hydroxides (heat treatment) = metal oxides. According to this process, we remove NaCl by washing the solution several times by distilled water. In the literature, metal oxides have been successfully produced by this method. We also did the same and published an experimental paper before. So, it works.
Is it relevant and interesting?
The approach is original. Predicting the possibility of obtaining powders with desired properties by the methods of thermodynamic calculations is interesting.
How original is the topic?
The topic, statement of research objectives and results are original.
What does it add to the subject area compared with other published material?
The originality of the approach is the key achievement of this work. The presented approach has drawbacks, because it is not tested for all oxides, but the approach itself is interesting.
Comment 2.3. Is the paper well written?
In general, the work is well written, however, the introduction should be revised in detail, regarding the remarks described above.
Reply 2.3. Thank you for your comment and suggestion. We revised introduction part in accordance with your suggestions.
Comment 2.4. Is the text clear and easy to read?
The text is difficult to read. English needs to be corrected.
Reply 2.4. Thank you for your comment. We did our best to improved the English of the manuscript.
Are the conclusions consistent with the evidence and arguments presented?
Yes, the conclusions to the work are consistent with the text of the work.
Do they address the main question posed?
Yes, the findings are consistent with the purpose of the study.